# Genomic and Environmental Factors Shape the Active Gill Bacterial Community of an Amazonian Teleost Holobiont

François-Étienne Sylvain,[a] Nicolas Leroux,[a] Éric Normandeau,[a] Aleicia Holland,[b] Sidki Bouslama,[a] Pierre-Luc Mercier,[a] Adalberto Luis Val,[c] Nicolas Derome[c]

[a]Institut de Biologie Intégrative et des Systèmes, Université Laval, Québec, Québec, Canada
[b]La Trobe University, School of Life Science, Department of Ecology, Environment and Evolution, Centre for Freshwater Ecosystems, Wodonga, Victoria, Australia
[c]Instituto Nacional de Pesquisas da Amazônia (INPA), Laboratório de Ecofisiologia e Evolução Molecular, Manaus, Amazonas, Brazil

**ABSTRACT** Fish bacterial communities provide functions critical for their host's survival in contrasting environments. These communities are sensitive to environmental-specific factors (i.e., physicochemical parameters, bacterioplankton), and host-specific factors (i.e., host genetic background). The relative contribution of these factors shaping Amazonian fish bacterial communities is largely unknown. Here, we investigated this topic by analyzing the gill bacterial communities of 240 wild flag cichlids (*Mesonauta festivus*) from 4 different populations (genetic clusters) distributed across 12 sites in 2 contrasting water types (ion-poor/acidic black water and ion-rich/circumneutral white water). Transcriptionally active gill bacterial communities were characterized by a 16S rRNA metabarcoding approach carried on RNA extractions. They were analyzed using comprehensive data sets from the hosts genetic background (Genotyping-By-Sequencing), the bacterioplankton (16S rRNA) and a set of 34 environmental parameters. Results show that the taxonomic structure of 16S rRNA gene transcripts libraries were significantly different between the 4 genetic clusters and also between the 2 water types. However, results suggest that the contribution of the host's genetic background was relatively weak in comparison to the environment-related factors in structuring the relative abundance of different active gill bacteria species. This finding was also confirmed by a mixed-effects modeling analysis, which indicated that the dissimilarity between the taxonomic structure of bacterioplanktonic communities possessed the best explicative power regarding the dissimilarity between gill bacterial communities' structure, while pairwise fixation indexes ($F_{ST}$) from the hosts' genetic data only had a weak explicative power. We discuss these results in terms of bacterial community assembly processes and flag cichlid fish ecology.

**IMPORTANCE** Host-associated microbial communities respond to factors specific to the host physiology, genetic backgrounds, and life history. However, these communities also show different degrees of sensitivity to environment-dependent factors, such as abiotic physico-chemical parameters and ecological interactions. The relative importance of host-versus environment-associated factors in shaping teleost bacterial communities is still understudied and is paramount for their conservation and aquaculture. Here, we studied the relative importance of host- and environment-associated factors structuring teleost bacterial communities using gill samples from a wild Amazonian teleost model (*Mesonauta festivus*) sampled in contrasting habitats along a 1500 km section of the Amazonian basin, thus ensuring high genetic diversity. Results showed that the contribution of the host's genetic background was weak compared to environment-related bacterioplanktonic communities in shaping gill bacterial assemblages, thereby suggesting that our understanding of teleost microbiome assembly could benefit from further studies focused on the ecological interplay between host-associated and free-living communities.

**KEYWORDS** 16S RNA, bacterioplankton, environmental microbiology, fish, genotype, gill, metagenomics, microbial ecology, population genetics, transcription

Address correspondence to François-Étienne Sylvain, francois-etienne.sylvain.1@ulaval.ca.

The authors declare no conflict of interest.

The Amazon River basin is one of the Earth's most diverse aquatic ecosystems. This region shows substantial spatial heterogeneity due to dramatic hydrochemical and ecological gradients that impose physiological constraints upon its aquatic communities (1–6). Essentially, 2 distinct water types, "white" and "black" waters, result from contrasting physicochemical profiles and characterize the Amazon River basin (7). White water environments are eutrophic (nutrient- and ion-rich with high productivity), have a circumneutral pH and typically are highly turbid (7–10). In contrast, black water environments are oligotrophic (nutrient- and ion-poor with low productivity) and contain a high concentration of dissolved organic carbon (DOC) enriched in organic acids, such as humic substances. The relatively high amounts of humic DOC acidify black water environments (pH 3.0 to 5.0), making them physiologically challenging for local fish (11). For instance, acidic and ion-poor water is known to affect the homeostasis of ionoregulatory processes in different ways (1): Animals living in hypo-osmotic environments generally face the osmotic influx of water and diffusive loss of salts to the external environment (12). In addition, when exposed to acidity (e.g., pH 3.5 to 5.0), the inhibition of active $Na^+$ and $Cl^-$ uptake and the elevation of passive ion loss is triggered, resulting in reduced plasma $Na^+$ and $Cl^-$ levels (13).

Previous studies have examined the effects of these water types on various aspects of Amazonian fish ecology, including phylogeography (14–16), migratory patterns (17), evolutionary strategies (18), and community diversity (19). However, to date only 1 investigation has studied the effects of black and white waters on native fish bacterial communities (20). Host-associated bacterial communities can often be seen as an extension of the host genome by providing functions critical for the host survival in changing or extreme environments (21). Together with the host (and other non-bacterial microbes), they constitute a holobiont, a term coined by Margulis & Fester (1991) (22), referring to the co-dependence between the host and its microbial symbionts. Measuring to what extent water types affect Amazonian fish bacterial communities is an important first step to understand the physiology of fish-microbe systems in the different Amazonian environments.

Amazonian fish bacterial communities may show a response to the different water types as water physicochemical parameters are generally known to influence the composition and expression of aquatic microbial communities (23, 24). For instance, a study on the Amazonian tambaqui (*Colossoma macropomum*) detected a high sensitivity of its external bacterial communities to slight variations in environmental physicochemical parameters (25). Furthermore, Amazonian fish bacterial communities could vary according to the differences in the environmental bacterioplanktonic communities, which also differ between water types (26). In other ecosystems, these communities have been known to interfere with host-associated bacterial assemblages via the production of bioactive compounds (27, 28), or by modulating the assembly process of the eukaryotic host symbiotic community (29–31).

In addition to environment-dependant factors, such as water types, Amazonian fish bacterial communities could also be influenced by their hosts' genetic backgrounds, which are known to affect the taxonomic profiles of fish bacterial communities in a variety of fish clades from other ecosystems, such as brook charr, Atlantic salmon, stickleback, elasmobranchs, whitefish, and Amazonian piranhas (32–37). However, the covariation between fish genotypes and the composition of their associated bacterial communities is not always clear, as some studies suggest that environmental influences overcome host genomic effects on fish bacterial communities, rendering the detection of the latter difficult (38, 39).

The flag cichlid (*Mesonauta festivus*) has recently emerged as an interesting model to study the factors shaping Amazonian fish bacterial communities (20, 40). The flag cichlid is a small (15 cm max), sedentary, detritivorous, and gregarious species that is abundant on the margins of almost all Amazonian lakes and rivers (41). This species is native to both black and white water environments, and thus enables the study of the effect of water type on fish bacterial communities. Furthermore, a recent analysis of

the flag cichlid phylogeography indicated that water types had a low predictive power on the genetic structuration observed in this species, which was better explained by a strong influence of past vicariant events (16). In their study, Leroux et al. (2022) (16) showed that flag cichlids were divided into 4 genetically differentiated populations (referred to as "genetic clusters" in this manuscript), 2 of them encompassing ecosystems of both black and white water (Fig. S1). Overall, this simple system composed of 4 main genetic clusters and 2 water types constitutes an interesting model to study the relative contribution of genomic and environmental factors shaping Amazonian fish bacterial assemblages.

Here, our study aimed to characterize how the gill bacterial community of wild flag cichlids differ among the water types and host genetic clusters. We analyzed a data set characterizing the gill bacterial communities of 240 flag cichlids distributed throughout 12 sampling sites in black and white water environments. These individuals were also used in the phylogenomic investigation of Leroux et al. (2022) (16) and, thus, represent the 4 genetic clusters previously identified in that study. We chose to study the gill bacterial communities, rather than those from the gut or the skin mucus, because the gill is the organ mostly responsible for ionoregulatory processes associated to the fish physiological response to different water types (reviewed in reference [11]). We used a 16S rRNA metabarcoding approach based on RNA extractions to characterize the transcriptionally "active" part of the gill bacterial communities. We quantified to what extent bacterial samples cluster according to their water type and their fish host's genetic cluster. Then, we identified bacterial biomarkers specific to the different water types and genetic clusters. Finally, we modeled the contribution of both environment-specific factors (i.e., free-living bacterioplankton taxonomic structure and environmental physicochemical parameters) and host-specific factors (i.e., Single Nucleotide Polymorphisms used to define the genetic clusters) on gill bacterial communities beta-diversity.

## RESULTS

After sequencing, merging, and quality filtering, 5,102,333 reads (mean of 21,620 reads/sample, N = 236 samples) were obtained for the gill bacteria data set. Gill bacterial communities of flag cichlids showed an increased relative abundance of 16S rRNA gene transcripts for Proteobacteria (mean of 65% $\pm$ 4%), Bacteroidetes (26% $\pm$ 4%), and Firmicutes (7% $\pm$ 2%) (Fig. 1). Firmicutes were significantly enriched in active gill bacterial communities of fish collected from black water (12% $\pm$ 5% in black water compared to 4% $\pm$ 1% in white water, $P = 0.06$). In contrast, Bacteroidetes were significantly enriched in active gill bacterial communities of fish collected from white water (34% $\pm$ 4% in white water compared to 15% $\pm$ 2% in black water, $P = 0.03$).

**Gill bacterial communities from different water types and host genetic clusters.** The principal coordinates analyses (PCoAs) shown in Fig. 2a indicate that, to a certain extent, gill bacteria samples clustered according to their genetic cluster of origin. The first 2 axes of these PCoAs represented 21.1% and 23.3% of the variance. The PERMANOVAs also suggested that the active gill bacterial communities significantly differed according to the genetic cluster of the host fish, both in black ($F_{3,97} = 7.49$, $R^2 = 13\%$, $P < 0.001$) and in white water ($F_{3,113} = 10.76$, $R^2 = 14\%$, $P < 0.001$) environments. Additionally, the PCoAs shown in Fig. 2b suggested that gill bacteria samples clustered according to their water type of origin. This result was obtained for the genetic clusters GC2 and GC4 for which we found samples in both black and white water types (this was not the case for the genetic clusters GC1 and GC4, thus they were excluded from this analysis). The first 2 axes of the 2 PCoAs in Fig. 2b represented 37% and 36% of the variance. PERMANOVAs confirmed that the active gill bacterial communities significantly differed according to the water type of the sampling environment, both in genetic clusters GC2 ($F_{2,57} = 14.30$, $R^2 = 20\%$, $P < 0.001$) and GC4 ($F_{2,37} = 9.33$, $R^2 = 20\%$, $P < 0.001$).

We identified bacterial biomarkers (at the amplicon sequence variance [ASV] level) specific to the 4 genetic clusters (Fig. 3a and b) or the 2 water types (Fig. 3c and d). We identified 90 bacterial biomarkers specific to one of the genetic clusters (23 ASVs for GC1, 9 for

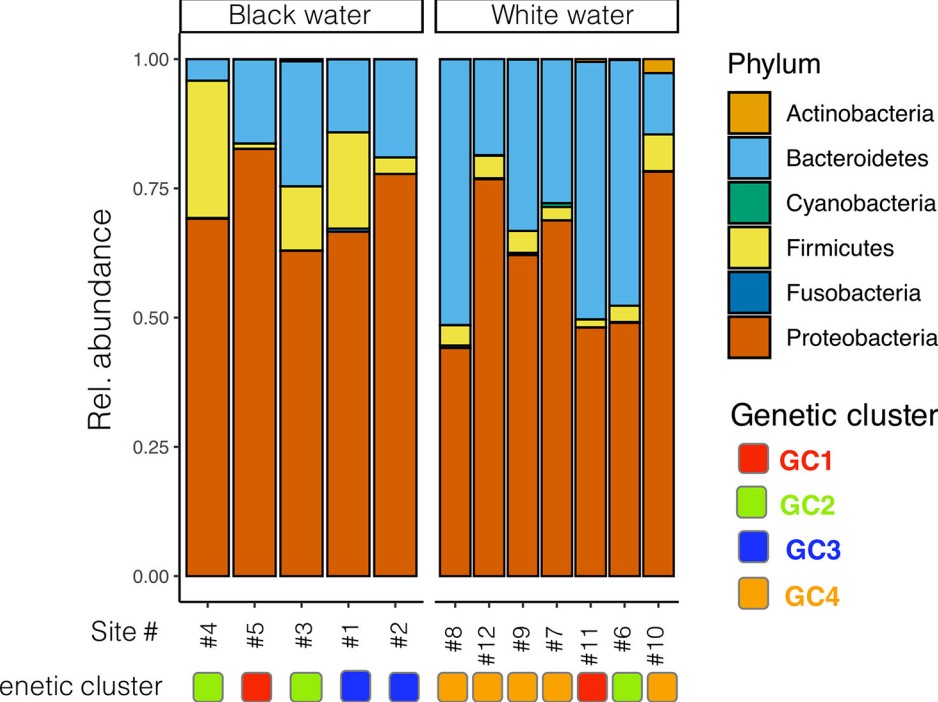

**FIG 1** Relative abundance of the phyla detected in active gill bacterial communities samples based on 16S rRNA gene transcripts, according to the water type and the genetic cluster of the fish found at each sampling site. "GC" stands for "Genetic cluster".

GC2, 29 for GC3, 29 for GC4). Their taxonomic assignations mostly represented members of the classes Alphaproteobacteria, Betaproteobacteria, Flavobacteriia, and Clostridia (Fig. 3b). Biomarkers from the Clostridia class were mostly associated to GC1 and GC2, those from the Flavobacteriia and Gammaproteobacteria classes were enriched in fish from GC3, while those from the Alphaproteobacteria and Betaproteobacteria class were found in fish from all genetic clusters (GC1, 2, 3, 4) but were mostly abundant in GC4. On average, these biomarkers represented a relative abundance of 13.8% of the global gill bacterial community (11.5% in GP1, 7.0% in GP2, 10.0% in GP3, 26.5% in GP4).

Concerning the bacterial biomarkers specific to water types, a total of 125 ASVs were identified: 69 ASVs associated to black water and 56 ASVs to white water flag cichlids. Biomarkers specific to individuals from black water were mostly from the Betaproteobacteria, Deltaproteobacteria, and Clostridia classes, while those specific to individuals from white water were characterized by a relatively higher abundance of Flavobacteriia and Gammaproteobacteria. The biomarkers specific to water types represented an average of 28.4% of gill bacterial communities (27.2% for black and 29.6% for white water fish).

**Gill and bacterioplankton communities in an environmental gradient.** After sequencing, merging, and quality filtering, 1,137,185 reads were obtained (mean of 16,973 reads/sample, N = 67 samples) for the bacterioplankton data set. Bacterioplankton samples showed a high relative abundance of 16S rRNA gene transcripts for Proteobacteria (mean of 57% ± 4%), Actinobacteria (24% ± 3%), and Firmicutes (17% ± 3%) (Fig. 4a). Proteobacteria were significantly enriched in black water environments (68% ± 7% in black water compared to 49% ± 4% in white water, $P = 0.02$), while Actinobacteria were significantly enriched in white water environments (29% ± 3% in white water compared to 18% ± 4% in black water, $P = 0.04$). A detailed characterization of these bacterioplankton communities can be found in Sylvain et al. (2021) (37).

The environmental parameters measured in this study and used to compute constrained analyses of principal coordinates (CAP) constrained ordinations (Fig. 4b and c)

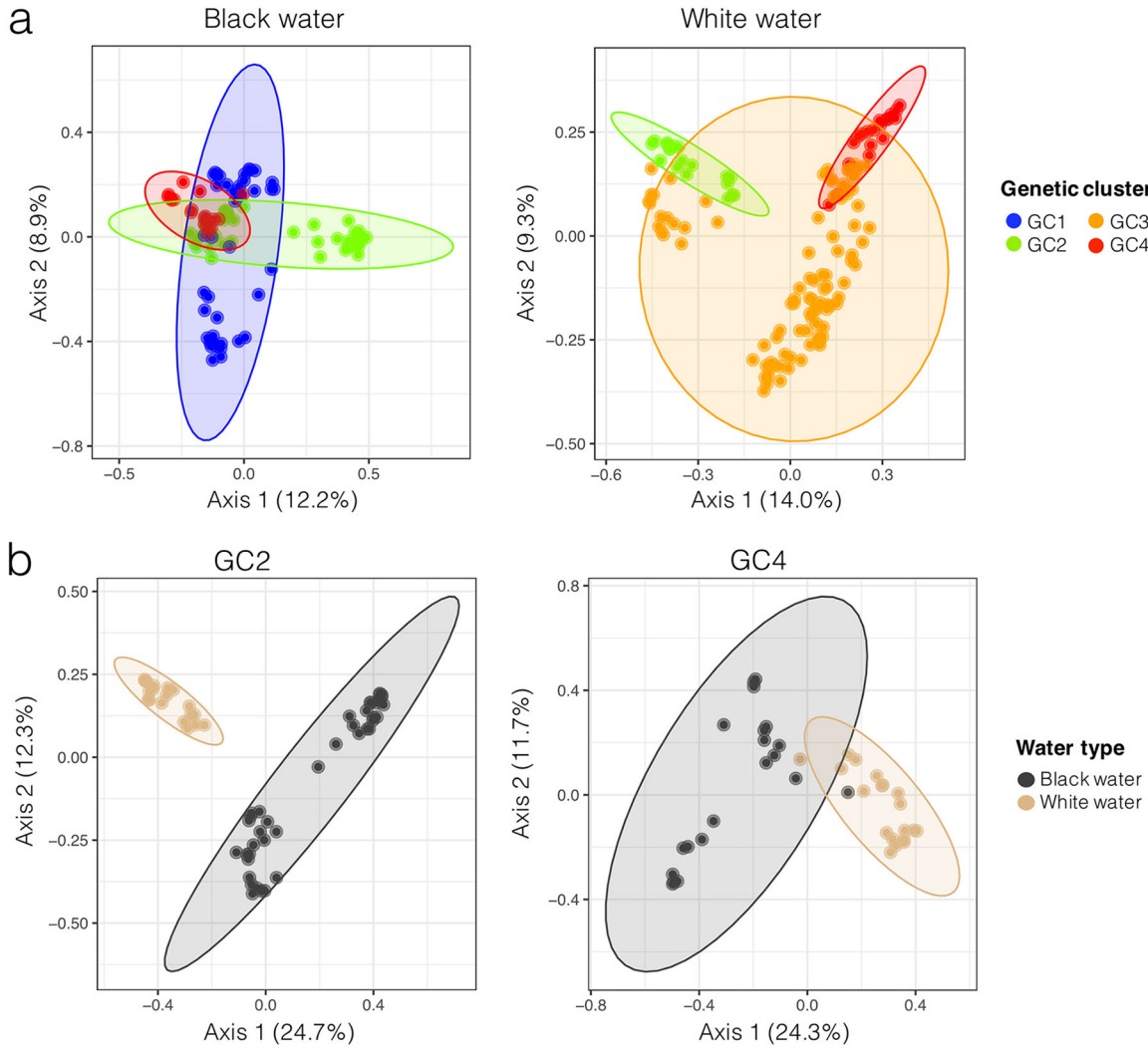

**FIG 2** Principal coordinates analyses (PCoA) of gill bacterial communities samples, based on Bray-Curtis distances. Ellipses represent groups with default confidence intervals of 0.05. In (a) samples are colored according to their genetic cluster of origin. "GC" stands for "Genetic cluster". In (b) samples are colored according to their water type of origin. The genetic clusters GC1 and GC3 are not shown in (b) as they were not found in both water types.

indicate that the 12 sampling sites have 2 contrasting physicochemical profiles that are typical of black and white water types (Tables S2, 3, 4, and 5, and Fig. S5). Additional physicochemical characterization of these sampling sites can be found in the Suppl. Mat. section "Environmental characterization" and in (37).

The CAP ordinations (Fig. 4b and c) revealed that samples cluster according to their water type of origin for gill and bacterioplankton communities. Permutation tests for CAP under the reduced model showed that the samples significantly clustered according to the 5 environmental variables provided for the constrained ordination (water conductivity and concentrations of DOC, aluminum, silicate, and chlorophyll a), both for bacterioplankton ($F_{2,61}$ = 2.53, $P < 0.001$) and gill bacterial communities ($F_{2,230}$ = 9.72, $P < 0.001$). The percentages of the variance retained by the first 2 dimensions of the CAP were relatively low: 10.51% for bacterioplankton and 10.82% for gill bacterial communities. This significant difference between communities from different water types was confirmed by a PERMANOVA (bacterioplankton: $F_{2,65}$ = 3.51, $P < 0.001$; gill bacteria: $F_{2,234}$ = 14.59, $P < 0.001$). Fitting of selected environmental variables on CAP (Fig. 4b and c) and *envfit* results (Table S6) indicated that variations in the concentrations of Al and DOC were significantly correlated to gill bacterial communities from

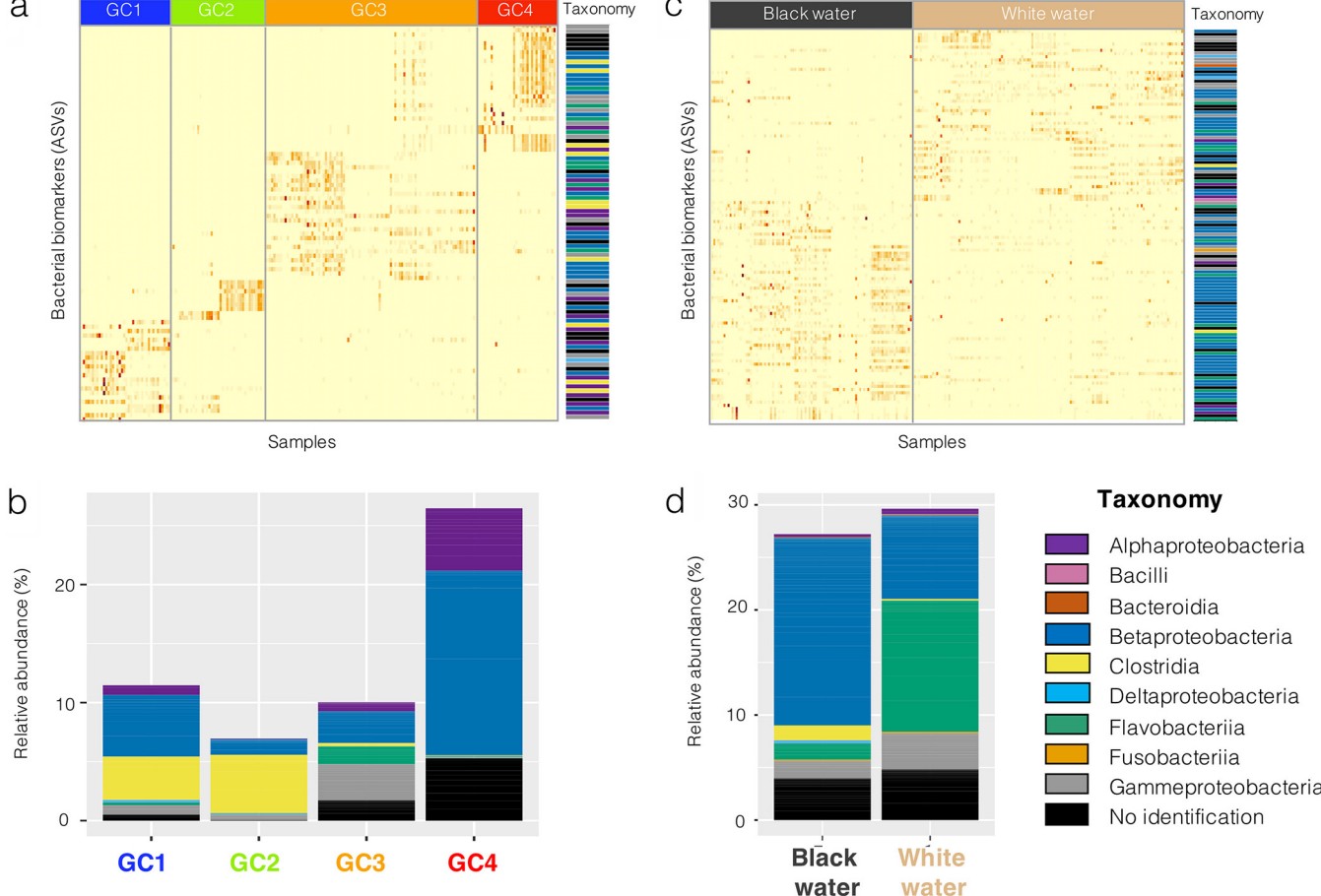

**FIG 3** Bacterial biomarkers (at the ASV level) specific to one of the four genetic clusters (a) and (b) or one of the two water types (c) and (d). In (a) and (c) the relative abundance of these biomarkers in each genetic cluster (a) or water type (c) is shown on heatmaps where each column is a gill bacterial community sample and each row represents one of the biomarker ASV (colors at the right end of each row represent biomarker phylogeny at the Class level). Darker shades of orange indicate higher relative abundance of the ASV in the sample. In (b) and (d) the relative abundance of the biomarkers in each genetic cluster (b) or water type (d) is shown on stacked barplots. In these plots, biomarker ASVs are colored according to their phylogeny (Class level).

black water sites (Al: $R^2 = 0.17$, $P < 0.001$; DOC: $R^2 = 0.09$, $P < 0.001$), while conductivity, concentration of chlorophyll a and silicates were associated with gill bacterial communities from white water sites (cond.: $R^2 = 0.17$, $P < 0.001$; chl. A: $R^2 = 0.23$, $P < 0.001$; silicate: $R^2 = 0.11$, $P < 0.001$). Interestingly, the bacterioplankton was not sensitive to the same physicochemical parameters, as the *envfit* analysis points out. Instead of being correlated with the 5 parameters above, planktonic communities were significantly correlated with several parameters associated with DOC characteristics, as well as the concentration of phaeopigments, Cr, Fe, and Cu (scores in Table S6). Gill bacterial samples displayed a more pronounced difference between water types than bacterioplankton samples, however, this could be due to the uneven number of samples (N gills = 236, N bacterioplankton = 67).

**Modeling environmental and genetic effects on gill bacterial community beta-diversity.** We used a linear mixed-effects modeling (LMER) approach to identify the main factors, or combination of factors, that explain the beta-diversity patterns (Bray-Curtis [BC] distances) observed on flag cichlid gill bacterial communities with the most precision and parsimony (Fig. 5). Spearman correlations between the factors considered in the LMER models varied between 0.004 and 0.31, the highest being between geographical distances and pairwise fixation indexes ($F_{ST}$) of fish from different sampling sites (Fig. S4). The dissimilarity values predicted by the global model (Fig. 5a), including all explanatory variables, were significantly correlated to the observed data (BC distances) from the fish gill bacterial

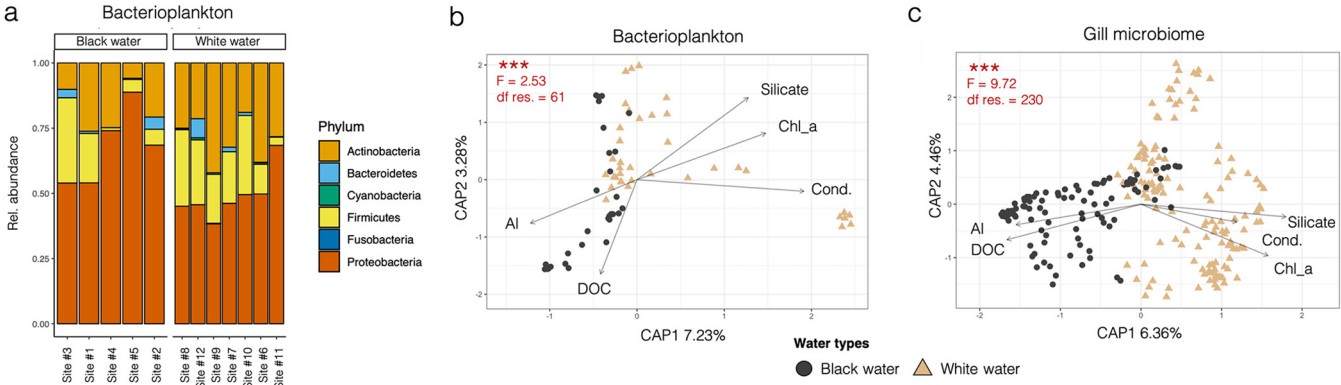

**FIG 4** (a) Relative abundance of the main phyla detected in bacterioplankton samples. In (b) and (c): Constrained analysis of principal coordinates (CAP) on bacterioplanktonic communities (b) and on gill bacterial community samples (c). Each data point in the CAP plots represents a sample, and their color and shape correspond to the water type at the sampling site. The results of permutation tests for CAP under reduced model (5 environmental variables) are shown in red in the upper left corner of each plot. "df res" stands for residual degrees of freedom and "***" for "$P$ value < 0.001". "Al" stands for dissolved aluminum ($\mu$g L$^{-1}$), "DOC" for the concentration of dissolved organic carbon (mg L$^{-1}$), "Silicate" for the concentration of silicates (mg L$^{-1}$), "Chl_a" for the concentration of chlorophyll a ($\mu$g L$^{-1}$), "Cond." for the conductivity ($\mu$S).

communities ($R^2$ = 39.8%, $F_{1,64}$ = 42.3, $P$ < 0.001). The 8 models with Akaike weights >0, including the null model, are shown in the diagram in Fig. 5b. Of these, only 3 significant models emerged, and they included 2 out of the 4 explanatory variables included in the global model: BC distance among bacterioplankton samples and $F_{ST}$ values between fish from different sites. The models that included the geographical distance between sampling sites and the environmental parameters (Euclidean distances) were not significant as they displayed Akaike weights inferior to the null model. The model with the best score was only composed of the bacterioplankton BC distance variable (Akaike weight = 0.51, K = 5, AICc = -161.15), largely surpassing the second-best model, which only included the $F_{ST}$ values (Akaike weight = 0.14, K = 5, AICc = -158.55). The models including the bacterioplankton BC distance variable (Fig. 5c) had a model-averaged estimate of 0.45 (confidence interval 0.12 to 0.79), more than three times the estimate for models including the $F_{ST}$ values (0.13 with confidence interval 0.02 to 0.24) (Fig. 5d). Linear correlations plotted separately, outside of the LMER model (Fig. 5c, d, e, and f), showed that of the 4 explanatory variables considered, the bacterioplankton BC distance variable had the strongest correlation with gill bacterial communities' dissimilarity (Fig. 5c) ($R^2$ = 9.5%, $F_{1,64}$ = 6.7, $P$ < 0.05), while Euclidean distances between environmental parameters had the weakest correlation with gill bacterial communities' dissimilarity (Fig. 5f) ($R^2$ = 1.0%, $F_{1,64}$ = 0.6, $P$ = 0.4). Overall, the mixed-effects linear modeling analysis suggested that bacterioplankton dissimilarity was much more correlated to gill bacterial assemblages' dissimilarity than the host $F_{ST}$ values, and the 2 other explicative variables considered.

## DISCUSSION

Fish-microbe systems evolve according to a complex mixture of environmental and host-specific factors (42). External bacterial communities of Amazonian fish are known to be especially sensitive to environmental factors, most notably to the composition of bacterioplankton (40). Several studies have investigated the Amazonian bacterioplankton composition, from the upper (20, 37, 43, 44) to the lower Amazon and its plume in the Atlantic Ocean (45–48). The bacterioplankton communities characterized in our study were in accordance with results from previous studies: Proteobacteria, Actinobacteria, Firmicutes, and Bacteroidetes were also the main bacterioplanktonic phyla identified by Toyama et al. (2016) (43) and Santos-Junior et al. (2017) (44). In addition, these 2 investigations reported the occurrence of Cyanobacteria (up to 24% in certain samples) and Planctomycetes (up to 8%) which were also detected but were not as abundant in our study. These differences could be due to the fact that sampling sites visited in these investigations were different from ours, and that our molecular approach was focused on the detection of transcriptionally-active taxa only.

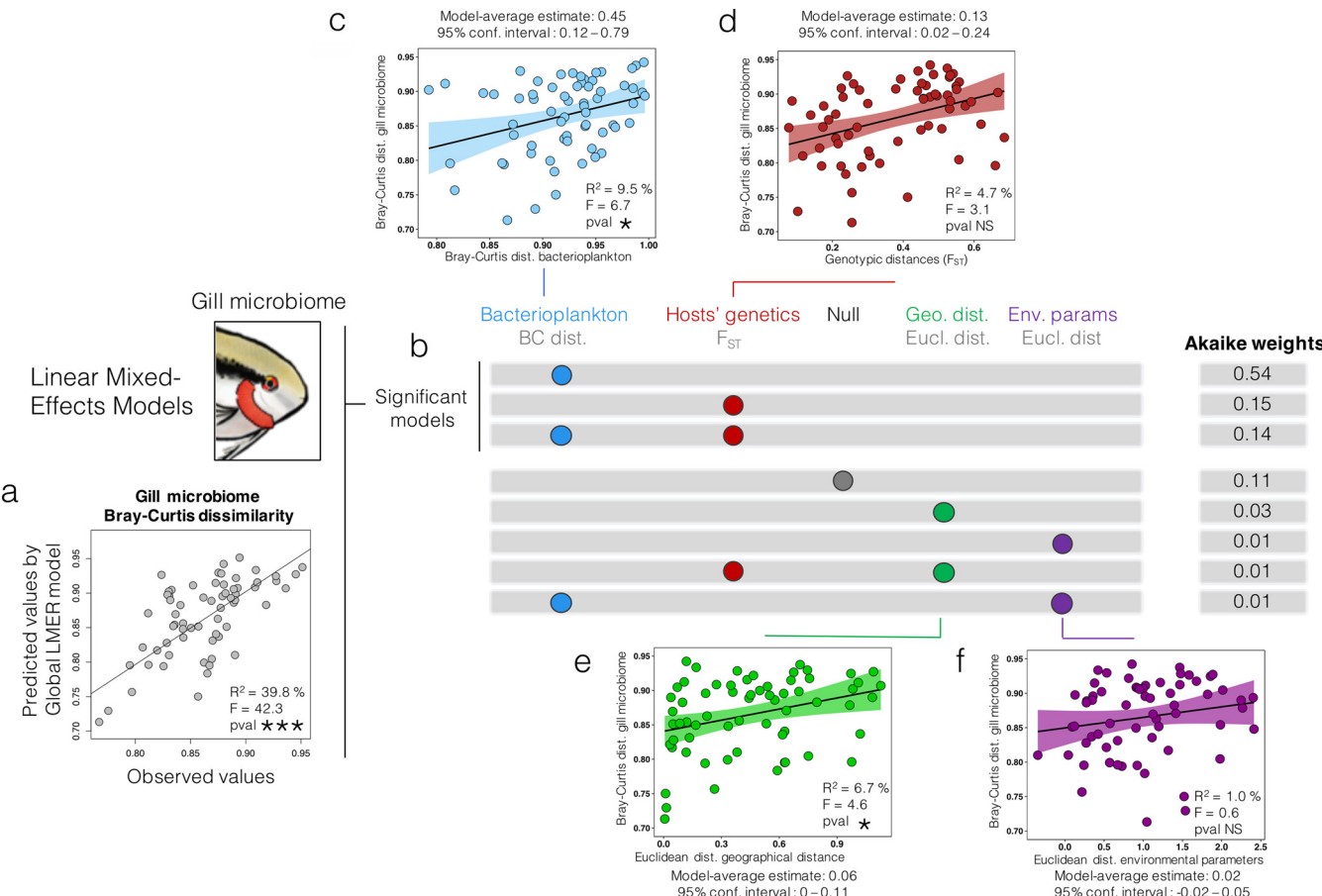

**FIG 5** Linear mixed-effects modeling analysis of the explicative variables potentially associated to gill bacterial community 16S rRNA gene transcripts. The 4 following explicative variables (transformed in distance matrices) are included in the global model (a) used for predicting dissimilarity values: bacterioplankton 16S rRNA gene transcripts (BC distance), genotypic distances between hosts ($F_{ST}$), geographic distances between sites (Euclidean distance), and the ensemble of the 34 environmental parameters measured in this study (Euclidean distance). Each row of the gray diagram in (b) corresponds to one of the models tested, and colored circles represent the elements that were included in the models. The upper three rows of the diagram represent significant models, with Akaike scores superior to the null (random) model (fourth row), while the lower four rows correspond to models with low explicative power. Linear correlation plots (c), (d), (e), and (f) for each of the 4 explicative variables considered in the model represent the linear correlation between the normalized distance/dissimilarity of each variable with the gill microbiome Bray-Curtis (BC) dissimilarity, when assessed separately outside of the LMER model. The linear correlations with BC gill bacterial community dissimilarity are displayed with the following variables: (c) BC bacterioplankton dissimilarity, (d) $F_{ST}$ genotypic distances, (e) Euclidean distances of geographical distances, (f) Euclidean distances of environmental parameters.

To our knowledge, this study is the first to characterize the gill bacterial community of an Amazonian fish. Several taxonomic groups detected on the gills were also found in the skin mucus of flag cichlids in a previous investigation (20). In both communities, Proteobacteria, Actinobacteria, and Bacteroidetes were among the most abundant phyla. Our results are also in accordance with a previous investigation describing the gill bacterial communities of 53 reef fish species, which identified Proteobacteria, Firmicutes, Fusobacteria, and Bacteroidetes as the most abundant phyla on fish gills (49). Furthermore, our results (Fig. 2b and Fig. 4c) show a strong response of the flag cichlid gill bacterial communities to the occupied water type. This response could depend on the different physicochemical conditions, characteristic of each water type, but also on the host physiological response to these conditions. For instance, a study on the gills of the Amazonian sardine (*Triportheus albus*) has shown a contrasted regulation of physiological processes in different water types (50). Some of these processes, such as the regulation of tight junction permeability, metal ion binding, and the host immune response (50), potentially involve gill bacterial communities (51–53).

The analysis of gill bacterial biomarkers has proven to be an essential tool in differentiating fish from contrasted environmental or health conditions (54). Here, the biomarkers' analysis (Fig. 3) highlighted the potential of several bacterial groups from the gills to

distinguish fish from different water types. Betaproteobacteria were mostly associated with black water fish (Fig. 3d). Interestingly, a recent study on Atlantic salmon showed a major enrichment of Betaproteobacteria in microbiomes characterized from whole gill samples instead of gill swabs (55). This suggests that the localization of selected members of the Betaproteobacterial group could be within cryptic tissue locations, such as beneath the surface epithelium. A few strains of Betaproteobacteria are known to be endosymbionts of fish gills (56, 57). Accordingly, in white water environments we detected several ASV biomarkers that were from the Flavobacteriia class, mostly represented by members of the genus *Chryseobacterium*, another clade known to include potential intracellular bacteria (58). In the future, using a correlative imaging approach, as done on mussel gills by Franke et al. (2021) (59), would help to resolve the potential niche partitioning of variable microbial communities across the ultrastructure of gill tissue and improve our understanding of the interaction between these water type biomarkers and their fish host in different habitats.

**Breaking down the factors driving active gill bacterial communities.** Not only the gills' bacteria (Fig. 4c), but also the bacterioplankton (Fig. 4b) and the environmental physicochemical parameters (Fig. S5) varied significantly according to water type. Gill bacterial communities are located at the frontier between the host and the external environment. This unique location enables them to interact extensively with their host and the metabolites of the bacterioplankton community, but it also exposes them to the variations in water physicochemical parameters. For this reason, we broke down the effect of water type into 2 factors in the LMER model: the environmental physicochemical parameters, and the bacterioplankton.

**Environmental physicochemical parameters.** It is well known that water physicochemical parameters drive, to a certain extent, the composition and expression of aquatic microbial communities (23, 24, 60). This has been highlighted previously in a study focused on Amazonian bacterioplankton (37). In our investigation, when all environmental physicochemical parameters were considered together to build distance matrices, the output was not significantly associated to patterns observed on gill bacterial communities (Fig. 4). However, when considered separately in CAP constrained ordinations, environmental physicochemical parameters did show significant correlation to gill community beta-diversity (Fig. 4c). Furthermore, results from the *envfit* analyses (Table S6) suggested a significant correlation between selected physicochemical parameters and the dissimilarity patterns observed for the gill bacterial communities. The results thus indicated that flag cichlid gill bacterial assemblages were sensitive to some (but not all) of the parameters associated with water type, most notably conductivity, Al concentrations, silicate, DOC, and chlorophyll a (Fig. 4c). Similarly, previous studies have observed a significant influence of conductivity (61), chlorophyll a (62), and DOC concentrations (40) on fish skin mucus bacterial communities.

**Bacterioplankton.** Among the factors considered in the modeling analysis, the bacterioplankton dissimilarity was the best explanatory variable, displaying the highest correlation with gill bacterial community dissimilarity (Fig. 5b and c). This result makes sense physiologically, as in this case, both the explanatory and the response variables were bacterial communities. Several studies have shown that microbial communities from different microhabitats (e.g., from different hosts sharing the same environment), can show similar responses when exposed to the same environmental conditions. For instance, studies on corals have shown that stressors such as lowered pH or eutrophication often lead to an increase in coral microbiome alpha diversity and a decrease in the bacterial symbiont *Endozoicomonas*, a pattern consistent across different coral species and geographical regions (reviewed in reference [63]). In our study system, it is possible that direct interactions exist between both bacterial community types considered here such that bacterioplanktonic bacteria could interfere with the activity of members from the gill bacterial community. For example, recent investigations suggested that the production of bioactive compounds, such as cyanotoxins produced by planktonic cyanobacterial strains, can significantly perturb fish microbiomes (27, 28).

An additional process linking planktonic and fish-associated bacterial communities relies on the assembly process of the later. Indeed, the assembly of bacterial assemblages on fish

largely depends on the past and present environmental pools of bacterial taxa to which the host was exposed (29–31). For instance, a study on Atlantic salmon has showed that the external skin mucus microbiome changes when the environment is modified: The bacterial community taxonomic structure of hatchery-raised juveniles converged with their wild counterparts shortly after transferring them from the hatchery to the wild (31). The importance of "source-sink" processes has also been shown in the bacterial communities' assembly of the Amazonian discus fish, where bacterioplanktonic taxa shape the gut community at early life stages, a period during which larvae bacterial assemblages show a weak resistance to colonization (30). The importance of source-sink processes might be even more important for gill communities, as they are exposed to environmental stressors, just like fish skin mucus communities. Such external communities are known to have a particularly dynamic composition that varies in accordance to fine-scale changes in environmental conditions (25, 64) and thus are more susceptible to frequent remodeling by environmentally-recruited taxa. Finally, while direct interactions between bacterioplankton and gill bacteria could be the cause of the correlation between these communities in our data sets, there is also a possibility that both communities show similar responses to shared water physicochemical parameters without interacting directly with one another. However, based on the close physical proximity of these communities and the high recruitment potential of environmental strains by fish bacterial communities, this perspective seems unlikely.

Overall, the high sensitivity of gill communities to water type could potentially be beneficial for fish holobionts in Amazonian ecosystems. Indeed, the rapid remodeling of gill communities could be an asset in heterogenous systems such as the Amazon basin, as it could facilitate the adaptation of fish to changing conditions, for instance during the important seasonal floods occurring in the flag cichlid's habitat during the wet season (January to July). Furthermore, since bacterial communities are known to play important roles in membrane permeability (reviewed in reference [65]), they are likely involved in the regulation of ionoregulatory processes occurring at the gills. In that case, a dynamic gill bacterial community that adapts rapidly to changing environmental conditions could be an essential tool to optimize ionic balance, especially in mixed-water environments receiving inputs of both ion-rich white waters and ion-poor black waters. Metatranscriptomic results from other Amazonian fish species would be needed in the future to test this hypothesis.

**The host's genetic effects.** The second-best model to explain the gill bacterial communities' beta-diversity patterns was the one including only the pairwise fixation indexes ($F_{ST}$) (Fig. 5b). Thus, a significant influence of the host's genetics on the gill bacterial community is not to be completely excluded, even though the best model only included bacterioplankton dissimilarity values. Indeed, the selection of the bacterioplankton model based on Akaike scores gave substantial weight to the parsimony criteria, thus advantaging simpler models (66, 67). Realistically, the global model makes the most sense biologically, as it included the influence of all the ecological, physicochemical and geographical explicative variables considered in this study, which are all known to affect the dynamics of host-associated microbiomes in many ecosystems (27, 37, 40, 68). It may seem surprising that the $F_{ST}$ values did not show a significant linear correlation with gill bacterial communities' beta-diversity (Fig. 5d), when this explicative variable alone constituted the second-best model (Fig. 5b). This result was likely due to the fact that the linear correlation in Fig. 5d did not account for the influence of the other explicative variables that also potentially influenced gill bacterial communities, while LMER models did. The LMER models thus enabled us to partition the effect of various variables acting simultaneously on gill communities.

Our study was one of the first investigations that combined bacterial community data and host's genetic information collected on the same fish specimens. Overall, the taxonomic structure of gill bacteria libraries was significantly different according to the host genetic cluster of origin (Fig. 2a), suggesting that the flag cichlid genetic background might influence the taxonomic structure of its bacterial communities. However, 3 main results suggest that the effect of the host genetics on gill bacterial assemblages was weak in comparison to

environmental factors. Firstly, Fig. 2b highlights that for the two genetic clusters that contained sites of both water types (GC2 and GC4), the differentiation between gill bacteria samples of different water types was stronger than between samples from different genetic clusters. The PERMANOVA tests resulted in higher F and $R^2$ values in Fig. 2b than in Fig. 2a, even though the number of residual degrees of freedom was lower for both genetic clusters in Fig. 2b. The percentage of variance explained by the first 2 axes of PCoAs in Fig. 2b were also higher than those from Fig. 2a. Secondly, the analysis of bacterial biomarkers (Fig. 3) revealed that biomarkers associated to water type constituted a higher relative proportion (125 ASVs representing an average of 28.4%) of the total bacterial community, than biomarkers associated to the genetic clusters (90 ASVs representing an average of 13.8% of the community). Thirdly, the relatively low Akaike score of the best LMER model including the host pairwise fixation index (i.e., three times inferior to the best one including bacterioplankton BC distance), as well as the relatively low model-average estimate of this factor (0.13) also suggest a weak correlation between host genetics and gill bacterial communities' taxonomic structure. A weak influence of host genetics on gill bacterial assemblages has also been reported in different genetic lines of rainbow trout (69). Additionally, a recent study focused on the external skin mucus microbiome of teleosts and elasmobranchs found that associations between hosts genetics and their microbiome composition was clade-specific (36), thus the patterns observed on the flag cichlid bacterial community might differ in other Amazonian fish.

**Conclusions.** This study on the flag cichlid active gill bacterial community is one of the first to include information on the hosts' genetic backgrounds, the free-living bacterioplanktonic pools of bacteria and environmental parameters. These data sets enabled us to study the relative contribution of genomic and environmental factors shaping the flag cichlid gill bacteria community in a simple natural system including host fish from four genetic clusters and 2 contrasting Amazonian water types. We observed that gill bacteria samples were significantly different between the host genetic clusters and between water types, suggesting a significant contribution of both host- and environment-specific factors in shaping these bacterial communities. However, constrained ordinations, PERMANOVAs, and analyses of bacterial biomarkers suggest that the contribution of the host's genetic background was relatively weak in comparison to environment-related factors in structuring the gill communities. This result was also confirmed by a mixed-effects modeling analysis, which suggested that the dissimilarity among bacterioplanktonic communities possessed the highest explicative power regarding the dissimilarity of gill bacterial communities, while pairwise fixation indexes ($F_{ST}$) from hosts only had a weak explicative power. Whether the relative importance of the host's genomic background on the gill microbiota is clade-specific should be confirmed in future studies targeting a broader range of Amazonian species and, if possible, using a reciprocal transplant experiment in controlled environments.

## MATERIALS AND METHODS

**Ethical approval.** This study was carried out in accordance with the recommendations of the Ethics Committee for the Use of Animals of the *Instituto Nacional de Pesquisas da Amazonia* (INPA). The permit (number 29837-18) was approved by the Ethics Committee for the Use of Animals of INPA.

**Fish sampling.** A total of 240 fish were collected from 12 sampling sites distributed throughout the upper Brazilian Amazon Basin in October-November 2018 and 2019 (dry season). Sampling fish during the dry season significantly facilitated the fish collection and enabled us to reach our objective of 20 specimens per sampling site. GPS coordinates and a map of all sites are found in Table S1 and Fig. 6a, respectively. Over the 12 sampling sites, there were 7 white water sites and 5 black water sites. Twenty fish specimens were collected at each sampling site using a combination of small seine net-fishing and line fishing. One gill arch was sampled from each specimen immediately after collection using sterile dissection tools (EtOH 70%). The biological samples were preserved in 2 mL of nucleic acid preservation (NAP) conservation buffer to preserve RNA integrity (70, 71), and were then stored at −80℃ until processing.

A fin clip was also sampled on the same specimens for the phylogenomic investigation of Leroux et al. (2022) (16). The genetic cluster (GC) of specimens sampled at each sampling site, previously determined by Leroux et al. (2022) (16), is detailed in Fig. 6, Fig. S1, and Table S1.

**Bacterioplankton sampling.** Six water samples were collected per site to characterize the bacterioplankton community. Surface water samples were taken at a depth of 30 cm in 2 L Nalgene (Thermo Fisher Scientific) bottles. Filtration was performed as in (72) through 22 $\mu$m-pore size polyethersulfone Sterivex filters (cat #SVGP01050, Millipore) less than 30 min after collection. Filters were also stored in 2 mL of NAP conservation buffer immediately upon collection, and then stored at −80℃ until

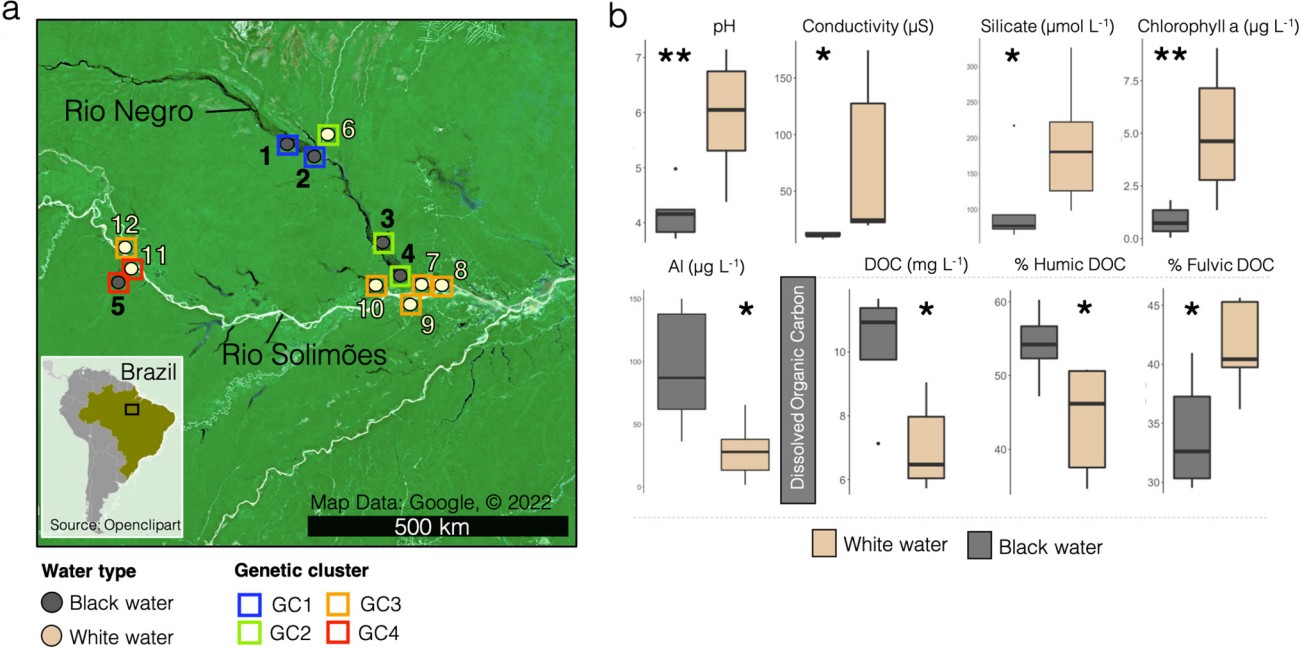

**FIG 6** (a) Map of the 12 sampling sites including information on the water type and the genetic cluster of fish found at each site. "GC" stands for "Genetic cluster". (b) Observed values of the environmental parameters commonly used to differentiate black from white water. "Al" refers to dissolved aluminum, "DOC" refers to dissolved organic carbon.

processing. Before RNA extraction, Sterivex filter casings were opened and processed according to (72) using sterile instruments, and filter membranes were stored in TRIzol (cat #15596026, Thermo Fisher Scientific).

**Characterization of environmental variables.** A total of 34 environmental variables commonly characterized in limnological studies (73) and associated to the physicochemical differences between the 2 water types (7) were measured (Tables S2, 3, 4, and 5). The methods used to measure the environmental variables are detailed in (37). Briefly, temperature (°C), conductivity ($\mu$S), pH, and dissolved oxygen (%) were measured directly at each site following fish and bacterioplankton sample collections, using a YSI professional plus series multimeter (YSI Inc/Xylem Inc). Two liter water samples were also collected at the same time, and transported on ice to the laboratory where the following parameters were immediately measured: the concentration of DOC, dissolved metals, nutrients, free ions, and chlorophyll a as described in (37). The water type (Table S1) of each site was determined based on the physicochemical profiles of the sampled environments (Fig. 6b and Tables S2, 3, 4, and 5). Our measurements of environmental parameters confirmed the *a priori* knowledge of the water types of these sampling sites.

**Biological sample processing. (i) Gills and bacterioplankton.** RNA extractions of whole gills and Sterivex filters (bacterioplankton) were performed according to the manufacturer's instructions of TRIzol without modification. Using RNA extracts prevented the detection of inactive, dead, or dormant taxa which are normally detected with a standard 16S approach based on DNA extracts. Four blank controls of NAP buffer were also processed identically to all samples for RNA extractions and sequencing. Microbial community taxonomic structure was assessed using a 16S rRNA approach conducted on RNA extracts. The RNA retrotranscription was performed with the qScript cDNA synthesis kit (cat #95048-100) from QuantaBio according to the manufacturer's instructions. Then, the fragment V3-V4 ($\sim$500 bp) of the 16S rRNA gene was amplified in cDNA extracts by PCR using the forward primer 347F (5'-GGAGGCAGCAGTRRGGAAT-3') and the reverse primer 803R (5'-CTACCRGGGTATCTAATCC-3') (74). The first PCRs of gill DNA were conducted in 25 $\mu$L according to the manufacturer's instructions of Q5 High-Fidelity DNA polymerase from New England BioLabs Inc (cat # M0491S) using an annealing temperature of 64°C and 35 amplification cycles. A PCR protocol with higher sensitivity was used for bacterioplankton samples due to their lower average cDNA concentration: PCRs of bacterioplankton cDNA were performed in 25 $\mu$L according to the manufacturer's instructions of the Qiagen Multiplex PCR kit (cat #206143) using an annealing temperature of 60°C and 30 amplification cycles. A second PCR was done using Q5 High-Fidelity DNA polymerase for both gill and bacterioplankton libraries (12 cycles), to add barcodes (indexes); more details in Suppl. Mat. section "Details on 16S rRNA library preparation". After each PCR, the amplified DNA of gills and bacterioplankton was purified with AMPure beads (cat #A63880, Beckman Coulter), according to the manufacturer's instructions, to eliminate primers, proteins, dimers, and phenols. Post-PCR cDNA concentrations were assessed on a Qubit instrument (Thermo Fisher Scientific) and by electrophoresis on 2% agarose gels. After purification, 16S rRNA amplicon libraries were randomly distributed among 4 Illumina MiSeq runs sequenced by the Plateforme d'Analyses Génomiques at the Institut de Biologie Intégrative et des Systèmes of Université Laval.

The R package *DADA2* (75) was used for ASV picking. Quality control of reads was done with the

*filterAndTrim* function using the following parameters: 290 for the forward read truncation length, 270 for the reverse read truncation length, 2 as the phred score threshold for total read removal, and a maximum expected error of 2 for forward reads and 3 for reverse reads. The filtered reads were then fed to the error rate learning, dereplication, merging, and ASV inference steps using the functions *learnErrors*, *derepFastq*, *mergePairs* and *DADA* from the *DADA2* pipeline (75). Chimeric sequences were removed using the *removeBimeraDenovo* function with the *consensus* method parameter. Taxonomic annotation of amplicon sequence variants (ASV) was performed by using *blastn* matches against NCBI "16S Microbial" database. As the NCBI database for 16S sequences is larger and updated more frequently than other sources, it provides more information about lesser-known taxa while minimizing ambiguous annotations. Matches above 99% identity were assigned the reported taxonomic identity (360 ASVs). Sequences with no match above the identity threshold (32,602 ASVs) were annotated using a lowest common ancestor method generated on the top 50 blastn matches obtained, a method inspired from the LCA algorithm implemented in MEGAN (76). Sequences with unassigned or non-bacterial taxonomic assignations were removed from the ASV table. Sequenced PCR negative controls were used to remove ASVs identified as potential cross contaminants using the *isContaminant* function from the *decontam* package with the default threshold of 0.4. Analyses of Shannon diversity according to sampling depth for each sample are provided in Fig. S1 and 2. ASV tables, metadata files, and taxonomy information were incorporated into *phyloseq*-type objects (77) before downstream analyses.

**(ii) Fin samples.** Detailed methods on the molecular approach, the construction of Genotyping-By-Sequencing libraries, sequencing and sequences' processing are found in Leroux et al. (2022) (16).

**Statistical analyses.** The statistical analyses describing the 4 genetic clusters of flag cichlids used in this study are detailed in (16), and their results are summarized in Fig. S1. Using stacked barplots (Fig. 1), we first visualized gill bacterial communities' taxonomic structure according to the water type and the genetic cluster of the host fish. We then used PCoA to study how the samples cluster according to the two aforementioned factors (Fig. 2). To compute PCoAs, we used a matrix of pairwise BC distances between all the gill bacteria samples (N = 240). The same BC distance matrix was used to compute permutational analyses of variance (PERMANOVA) to assess if gill bacteria samples significantly differed according to their water type or the genetic cluster of the host fish (Fig. 2). PERMANOVAs were computed with the *adonis* function from the *vegan* R package (78) using 1000 permutations.

Secondly, we identified bacterial biomarkers (at the ASV level) from the gill bacterial assemblage, that were significant discriminant features of host genetic cluster (Fig. 3a and b) or water type (Fig. 3b and c). To do so, we used the function *multipatt* from the *IndicSpecies* package (79) to identify discriminant ASVs based on their read abundance in the gill community ASV table. The ASVs which showed indicator values >0.5 and $P$-values < 0.01 after 999 permutations were retained as discriminant features for 1 of the groups from the comparison (i.e., the 4 genetic clusters or the 2 water types). The indicator value used for biomarker identification incorporates a correction for unequal group sizes (80). The relative abundance of these biomarkers was represented in heatmaps (Fig. 3a and c) and stacked barplots (Fig. 3b and d).

Thirdly, we studied in more detail how the taxonomic structure of gill bacterial communities varied according to two environment-related factors: Bacterioplankton taxonomic structure and environmental parameters (Fig. 4). To do so, we first represented the relative abundances of transcripts from the different bacterial phyla detected in bacterioplankton communities using stacked barplots (Fig. 4a). Then, to visualize how samples cluster according to their water type of origin, we used the *capscale* function from *vegan* R package (78) to compute CAP. This analysis showed the variation in the bacterioplankton or gill bacterial communities (response variables) explained by environmental parameters (explanatory variables) (Fig. 4b and c). In brief, CAP is a distance-based ordination method similar to redundancy analysis (81), but it allows non-Euclidean dissimilarity indices such as the BC distance. The R function *capscale* is based on Legendre and Anderson (1999) (81): the dissimilarity data is first ordinated using metric scaling, and the ordination results are then analyzed using a redundancy analysis (RDA). BC distance was used for the computation of CAPs. Covariation and multicollinearity were assessed by measuring variance inflation factors (VIF) (82) using the function *vif.cca*. Only environmental parameters not significantly correlated to each other, with VIF scores < 10 were retained as explanatory variables. Consequently, the 5 following parameters were retained for this analysis: water conductivity ($\mu$S/L), and the concentrations of DOC (mg/L), aluminum ($\mu$g/L), silicate ($\mu$mol/L), and chlorophyll a ($\mu$g/L) (Fig. 4b and c). Permutation tests (999 permutations) for CAP under the reduced model (the 5 aforementioned environmental variables) were conducted by running *anova* on the *capscale* result. The *envfit* function from *vegan* (78) was used to measure the strength of association between environmental variables and bacterial community data (from the relative abundance table of ASVs). To do so, *envfit* fitted the environmental vectors (water conductivity, and concentrations of DOC, Al, silicates, and chlorophyll a) onto the distance-based CAP ordination. PERMANOVAs based on BC distances were computed to detect significant differences between samples of different water types.

Finally, we constructed LMER models from ecological, physio-chemical, and geographical variables, that could potentially affect the gill communities' structure characterized in this study (Fig. 5). These factors included: the bacterioplankton taxonomic structure, host fish genetics, geographical distance between sampling sites, and the environmental parameters measured. Here, the factor "Hosts' genetics" does not refer to the 4 genetic clusters considered in Fig. 1, 2, and 3, but rather to the list of single nucleotide polymorphisms (SNP) used to compute pairwise fixation indexes ($F_{ST}$) with the LMER model. Details on the computation of $F_{ST}$ values are found in Leroux et al. (2022) (16). The covariation between these four factors was assessed using Spearman correlations, before model construction (Fig. S4). Our LMER model was based on distance matrices, thus the following distances were calculated between samples of different sampling sites: (i) BC distance for bacterioplankton and gill bacterial 16S rRNA gene

transcripts' libraries; (ii) $F_{ST}$ were estimated for host fish genetic dissimilarity; (iii) Euclidean distances were computed (after normalizations based on the mean and standard deviation) for geographical distances, and for the set of environmental parameters. We identified the combination of factors which most likely and parsimoniously explain the patterns observed in the beta-diversity (BC distance) of gill bacteria transcripts based on Akaike weights, using the model selection tool *aictab* from the R package *AICcmodavg* (83, 84). A global model containing all potential factors (Fig. 5a) and a null model (random vector) were also included in the analysis. The fitting of the models to gill community data was done with *lmer* from *lme4* on R (67). Model-averaged parameter estimates, the unconditional standard error and unconditional confidence intervals were computed using *AICcmodavg* (83, 84). Linear correlation of individual variables and the Bray-Curtis distances of the gill bacterial communities were also plotted separately (Fig. 5c, d, e, and f), outside of the LMER models.

**Data availability.** The 16S rRNA gene transcripts' data sets generated and analyzed during this study can be found in the Sequence Read Archive (SRA) repository, BioProjectIDs: PRJNA839167 and PRJNA839174 (data from gills), PRJNA736442 and PRJNA736450 (data from bacterioplankton). The R scripts used for the 16S RNA sequence analysis and the input files including all metadata and ASV taxonomy are freely available on the Open Science Network platform (URL: https://osf.io/d3n76/).

## SUPPLEMENTAL MATERIAL

Supplemental material is available online only.

**SUPPLEMENTAL FILE 1**, PDF file, 0.8 MB.

## ACKNOWLEDGMENTS

We thank the National Geographic Society, *NSERC*, MITACS, and Ressources Aquatiques Québec for awarding travel and field work grants to FÉS. This study was part of the NSERC Discovery grant of N.D., the INCT ADAPTA project of A.L.V., and supported by a Canada-Brazil Awards – Joint Research Project of N.D. and A.L.V., by CNPq, FAPEAM and CAPES.

We thank Thiago Nascimento, Reginaldo Oliveira, and Nazaré Paula for technical support with field work logistics. We thank Roxanne Dhommée for support in the molecular biology laboratory work. Finally, we thank the anonymous reviewers that generously took the time to help improve this manuscript.

We declare that there are no conflicts of interest.

F.-É.S., A.L.V., and N.D. designed the study; F.-É.S., N.L., A.H., and N.D. performed field sampling; F.-É.S., N.L., and P.L.M. conducted RNA extractions and prepared 16S rRNA libraries; F.-É.S., N.L., and S.B. conducted the bioinformatical and statistical analyses; F.-É.S. wrote the manuscript; all authors revised the manuscript.

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
