## [Reviewer comments · Microbiology Spectrum]

Microbiology Spectrum

Genomic and environmental factors shape the active gill bacterial community of an Amazonian teleost holobiont

Francois-Etienne Sylvain, Nicolas Leroux, Eric Normandeau, Aleicia Holland, Sidki Bouslama, Pierre-Luc Mercier, Adalberto Luis Val, and Nicolas Derome

Corresponding Author(s): Francois-Etienne Sylvain, Laval University

Review Timeline:

Submission Date:	June 2, 2022
Editorial Decision:	August 9, 2022
Revision Received:	November 8, 2022
Accepted:	November 11, 2022

Editor: Blaire Steven

Reviewer(s): The reviewers have opted to remain anonymous.

Transaction Report:

DOI: <https://doi.org/10.1128/spectrum.02064-22>

August 9, 2022

Mx. Francois-Etienne Sylvain
Laval University
Biology
1030 avenue de la Médecine
Quebec, QC
Canada

Re: Spectrum02064-22 (Genomic and environmental factors shape gill microbiome activity in an Amazonian teleost holobiont)

Dear Mx. Francois-Etienne Sylvain:

Link Not Available

Sincerely,

Blaire Steven

Journals Department
Reviewer comments:

Reviewer #1 (Comments for the Author):

This manuscript is very interesting, with robust data being presented (but see my comments on sampling), appropriate analysis and conclusions. I have a few comments which I think are worth addressing before publication. The first is conceptual. Authors talk about microbiome activity throughout the text which implies some sort of analysis of microbiome function. In fact they performed metataxonomic analysis of active bacteria, and identified "bacterial biomarkers specific to different water types and genetic clusters" and modelled the contribution of bacterioplankton, environmental variables and host genetics towards bacterial beta diversity. Only bacteria were targeted (not the microbiome) and they still used the traditional metataxonomic (metabarcoding) approach. In my view researchers should be more careful when describing their data (in this case 16S rRNA

data) and should follow the most recent recommendations in this regard (just as an example see Berg et al (2020). *Microbiome*. 8:103: "The term microbiome, as it was originally postulated by Whipps and coworkers, includes not only the community of the microorganisms, but also their "theatre of activity." The latter involves the whole spectrum of molecules produced by the microorganisms, including their structural elements (nucleic acids, proteins, lipids, polysaccharides), metabolites (signaling molecules, toxins, organic, and inorganic molecules), and molecules produced by coexisting hosts and structured by the surrounding environmental conditions."). One of the strengths of this work is that RNA, and therefore active microbiota, was used as starting material. This is an excellent approach but since only a portion of the bacterial 16S rRNA gene was amplified, what is being effectively measured are active bacterial abundances and there is no attempt in defining what is their activity or function (e.g. by sequencing whole transcripts). The authors could have tried to use metataxonomic data to infer on metabolic potential of the communities. Although results would have to be interpreted carefully however it would be very interesting to see the outcomes given the aims of the work. Sequencing whole RNA would have of course provided remarkable insights into the bacteriome and this could have taken the manuscript to a very different level. I thus encourage the authors to modify the title and text to accurately describe their aims and add a Picrust analysis or something similar.

Another important aspect is whether environmental variables were measured in the same day and place as fish and water collections? This needs elaboration in the text, i.e. they should be taken concomitantly and much more so when the authors examine active bacteria.

- were all samples sequenced together in a single MiSeq run? If not, were potential biases assessed?

- Taxonomic inference was made against NCBI "16S Microbial" database. Being aware of the innumerable misidentified genetic sequences in NCBI Genbank I wonder how are microbial taxonomic annotations affected by this and whether the database is actively curated to avoid these issues. In any case I think the way authors handled unidentified bacteria is excellent, but these results are missing (e.g. how many were identified using phylogenetics, etc). Also how did they deal with contaminants (e.g. fish sequences, non-bacteria) needs to be stated.

- It would be useful to have more details regarding the statistics behind the capscale and envfit functions to fully understand what they add to the results LMER model. This is more or less explained in Discussion but I think it deserves further attention in Methods.

minor comments:

- sometimes *bêta*-diversity is used instead of beta-diversity

Reviewer #2 (Comments for the Author):

In this study by François-Étienne et al, the authors examined the impact of host genotype and environmental factors on the gill microbiome of an Amazonian fish species, *Mesonauta festivus*. The study is really cool and worthwhile, the references cited are appropriate, and the methods used are correct (and thorough!). My main concern is with the writing. The grammar needs to be thoroughly revised throughout the manuscript. In addition, there were some odd word choices that did not really convey what I think the authors were trying for. I gave some examples of recommended changes, especially in the Introduction. I recommend the authors ask a native English speaker to help revise the manuscript. Finally, check the order in which figures appear in the text. The first figure mentioned is Fig. 2.

Introduction - You mention bacterioplankton only once in the Introduction, despite it being a major component (if not the main component) of your Results and Discussion. I think you need to better introduce and elaborate on the concept. At the very least, you should mention how water type impacts bacterioplankton communities, or discuss what is known about the impact of bacterioplankton on gill microbiomes.

Line 28 - "distributed across 12"

Line 30 - I'm not sure "transcriptionally active" is necessary here. Are there environments on fish where the microbiomes aren't transcriptionally active?

Line 32 - Replace "in light of" with "using"

Lines 34-36 - How were they different? Which clusters and treatments were more active?

Line 64 - Replace "diversified" with "diverse"

Line 65 - Remove "existence of"

Line 70 - Replace "are characterizing" with "characterize"

Line 72 - Replace "show a high turbidity" with "are highly turbid"

Line 77 - What do you mean by "physiologically challenging?"

Line 82 - Replace "evolutive" with "evolutionary"

Line 83 - Is "global community diversity" the right phrase here? The referenced study focuses on the Amazon.

Lines 90-92 - Some clarification is needed here. Do you mean that the transcriptomes of the microbes are sensitive to variations in environmental parameters, or that the microbial communities are in general sensitive to these variations? If it's the later, it's probably not necessary to specifically compare with the host fish transcriptome.

Line 107 - Replace "detritivore" with "detritivorous"

Line 109 - Replace "They are" with "The species is" or something along those lines. Otherwise, you are alternating between singular and plural.

Line 110 - It allows you to test the specific effect of water type, not environmental factors in general.

Lines 121-122 - Replace "gill microbiome of wild flag cichlids changes according to the environmental water type and the host genetic cluster" with "gill microbiomes of wild flag cichlids differ among the water types and host genetic clusters"

Line 133 - "specific to the different water types"

Line 136 - Remove "list of"

Line 142 - What do you mean by "important"? Do you mean increased?

Line 143 - Fig. 1 should be listed first. What you refer to as Fig. 1 is in the Methods section which is at the end of the manuscript.

Lines 144-147 - Reword these sentences. Rather say "Firmicutes were significantly enriched in the gill microbiomes of fish collected from black water..." and so on. Also, I would make it more explicit that you are comparing the black and white water microbiomes when you list the statistics, e.g., "12 {plus minus} 5 % in black water compared with 4 {plus minus} 1 % in white water; $p = 0.06$ ".

Line 152 - Remove "results from"

Lines 154-155 - When reporting an F-statistic, you should also report both the numerator and denominator degrees of freedom. One way in which F-statistics are commonly reported is as follows: " $F_{n,97} = 7.49, p < 0.001, R^2 = 13\%$ " with the degrees of freedom following the F in subscript. See Bowers et al. (2021) for a good example of how to report F-stats.

Line 166 (and throughout) - I would refer to these as taxa rather than biomarkers.

Line 208 - What were the five environmental variables?

Line 285 - Remove "one of the"

Lines 317 - Replace "decomposing" with "breaking down", or another synonym.

Fig. 3 - The figure caption needs more details. What do the ovals indicate? Also, the figure text is hard to read and needs to be larger. The statistics shown in the boxes are also listed in the main text, so may not need to be reported in both locations.

Fig. 4 - This figure is a bit confusing. Fig. 4a and c need a legend. It might be interesting to show how the "biomarkers" in 'a' and 'c' match up with the bar plots in 'b' and 'd' if possible.

Bowers, C., Toews, M. D., and Schmidt, J. M. 2021. Winter cover crops shape early-season predator communities and trophic interactions. *Ecosphere* 12(7):e03635

Staff Comments:

Preparing Revision Guidelines

Please return the manuscript within 60 days; if you cannot complete the modification within this time period, please contact me. If you do not wish to modify the manuscript and prefer to submit it to another journal, please notify me of your decision immediately so that the manuscript may be formally withdrawn from consideration by Microbiology Spectrum.

Response to reviewers – Spectrum02064-22R1

Reviewer #1 (Comments for the Author):

Comment: This manuscript is very interesting, with robust data being presented (but see my comments on sampling), appropriate analysis and conclusions.

Response: Thank you for this comment!

Comment: I have a few comments which I think are worth addressing before publication. The first is conceptual. Authors talk about microbiome activity throughout the text which implies some sort of analysis of microbiome function. In fact they performed metataxonomic analysis of active bacteria, and identified "bacterial biomarkers specific to different water types and genetic clusters" and modelled the contribution of bacterioplankton, environmental variables and host genetics towards bacterial beta diversity. Only bacteria were targeted (not the microbiome) and they still used the traditional metataxonomic (metabarcoding) approach. In my view researchers should be more careful when describing their data (in this case 16S rRNA data) and should follow the most recent recommendations in this regard (just as an example see Berg et al (2020). *Microbiome*. 8:103: "The term microbiome, as it was originally postulated by Whipps and coworkers, includes not only the community of the microorganisms, but also their "theatre of activity." The latter involves the whole spectrum of molecules produced by the microorganisms, including their structural elements (nucleic acids, proteins, lipids, polysaccharides), metabolites (signaling molecules, toxins, organic, and inorganic molecules), and molecules produced by coexisting hosts and structured by the surrounding environmental conditions.").

Response: The reviewer is absolutely right: Our study focused on bacteria and not on all components of the microbiome as defined by Whipps and coworkers. In the revised manuscript, we used the term "bacterial community" instead of "microbiome" at all relevant instances, to make sure to be as clear as possible.

One of the strengths of this work is that RNA, and therefore active microbiota, was used as starting material. This is an excellent approach but since only a portion of the bacterial 16S rRNA gene was amplified, what is being effectively measured are active bacterial abundances and there is no attempt in defining what is their activity or function (e.g. by sequencing whole transcripts). The authors could have tried to use metataxonomic data to infer on metabolic potential of the communities. Although results would have to be interpreted carefully however it would be very interesting to see the outcomes given the aims of the work. Sequencing whole RNA would have of course provided remarkable insights into the bacteriome and this could have taken the manuscript to a very different level. I thus encourage the authors to modify the title and text to accurately describe their aims and add a Picrust analysis or something similar.

Response: We definitely agree with the reviewer that studying the Amazonian fish bacterial transcriptomes at a functional level is interesting and important. In fact, our research group has already sequenced the metatranscriptomes (dual RNA-Seq of the whole RNA) of Amazonian host fish and their bacterial communities. This functional data comes from the same specimens of flag cichlids used in the current study and is analyzed in parallel with the metatranscriptomes of three additional fish species. The data is explored in another manuscript, which is currently under review (please see bioRxiv pre-print here <https://www.biorxiv.org/content/10.1101/2022.10.22.513327v1>). Thus, to prevent the dual

publication of the same results, we prefer to focus the current manuscript on the taxonomic structure of the “active” bacterial community (16S rRNA approach), and publish the functional analyses separately.

This being said, even though we used RNA as starting material, we agree with the reviewer that the current manuscript should make it clear that the data analyzed does not consist in full transcriptomes, but rather in 16S rRNA transcript libraries profiling the taxonomic structure of the “active” bacterial community. To clarify this aspect, we changed the formulation “gill bacterial community transcriptional activity” to “active gill bacterial community taxonomic structure” at every instance in the revised manuscript. Furthermore, we revised the title accordingly to change the formulation “gill bacterial activity” to “active gill bacterial community”.

Comment: Another important aspect is whether environmental variables were measured in the same day and place as fish and water collections? This needs elaboration in the text, i.e. they should be taken concomitantly and much more so when the authors examine active bacteria.

Response: The following specifications were added in the Methods of the revised manuscript (lines 519-525):

“Briefly, temperature (°C), conductivity (μS), pH and dissolved oxygen (%) were measured directly at each site following fish and bacterioplankton sample collections, using a YSI professional plus series multimeter (YSI Inc/Xylem Inc, Yellow Springs (OH), USA). Two liter water samples were also collected at the same time, and transported on ice to the laboratory where the following parameters were immediately measured: the concentration of DOC, dissolved metals, nutrients, free ions and chlorophyll a as described in (37).”

Comment: Were all samples sequenced together in a single MiSeq run? If not, were potential biases assessed?

Response: The samples were randomly distributed among four MiSeq runs. The following information has been added to clarify this aspect in the Methods section (lines 556-559):

“After purification, 16S rRNA amplicon libraries were randomly distributed among four Illumina MiSeq runs sequenced by the Plateforme d’Analyses Génomiques at the Institut de Biologie Intégrative et des Systèmes of Université Laval.”

Since they were randomly distributed, there were samples from every sampling site, water color and genetic cluster in each run. In that case, potential run-specific bias would not interfere with the genetic or environmental factors studied here.

Comment: Taxonomic inference was made against NCBI "16S Microbial" database. Being aware of the innumerable misidentified genetic sequences in NCBI Genbank I wonder how are microbial taxonomic annotations affected by this and whether the database is actively curated to avoid these issues. In any case I think the way authors handled unidentified bacteria is excellent, but these results are missing (e.g. how many were identified using phylogenetics, etc). Also how did they deal with contaminants (e.g. fish sequences, non-bacteria) needs to be stated.

Response: The reviewer is right, and this has been a topic of preoccupation for us during our investigation. Other databases such as SILVA and RDP were considered, however, even those databases have their own issues with mis-assigned taxonomic entries (Edgar 2018). The NCBI 16S Refseq record, like any other 16S database, possesses the same kind of risk, while offering

a more up-to-date collection of taxonomies. We chose to conduct the taxonomic identification using multiple entries through the LCA algorithm. Assuming that erroneous taxonomic annotations are minority events, their impact is negated during taxonomic assignment. Specific taxa of interest discussed in the paper were manually verified by the authors.

Overall, 360 ASVs (out of 32,962 ASVs retained post-filtration) were identified using NCBI “16S Microbial” database, while 32,602 ASVs were identified using the LCA method. The following information has been added to the revised manuscript (lines 572-576):

“Matches above 99% identity were assigned the reported taxonomic identity (360 ASVs). Sequences with no match above the identity threshold (32,602 ASVs) were annotated using a lowest common ancestor method generated on the top 50 blastn matches obtained, a method inspired from the LCA algorithm implemented in MEGAN (76).”

As previously mentioned, taxonomic annotation of ASVs was performed by using *blastn* matches against NCBI “16S Microbial” database. Thus, non-bacterial ASVs had unassigned taxonomic annotation. These sequences only represented 0.23 % (77 out of 32,962 ASVs) of the dataset, and were removed from the dataset before conducting downstream statistical analyses. In addition, potential bacterial contaminants were removed using the *isContaminant* function from the *decontam* package. This information is specified in the revised manuscript (lines 576-579):

“Sequences with unassigned or non-bacterial taxonomic assignments were removed from the ASV table. Sequenced PCR negative controls were used to remove ASVs identified as potential cross contaminants using the isContaminant function from the decontam package with the default threshold of 0.4.”

Reference

Edgar, R. 2018. Taxonomy annotation and guide tree errors in 16S rRNA databases. PeerJ. 5:e5030. doi: 10.7717/peerj.5030

Comment: It would be useful to have more details regarding the statistics behind the *capscale* and *envfit* functions to fully understand what they add to the results LMER model. This is more or less explained in Discussion but I think it deserves further attention in Methods.

Response: The following information on *capscale* was added (lines 623-632):

“Then, to visualize how samples cluster according to their water type of origin, we used the capscale function from vegan R package (78) to compute constrained analyses of principal coordinates (CAP). This analysis showed the variation in the bacterioplankton or gill bacterial communities (response variables) explained by environmental parameters (explanatory variables) (Fig. 4b,c). In brief, CAP is a distance-based ordination method similar to redundancy analysis (81), but it allows non-Euclidean dissimilarity indices such as the BC distance. The R function capscale is based on Legendre and Anderson (1999) (81): the dissimilarity data is first ordinated using metric scaling, and the ordination results are then analysed using a redundancy analysis (RDA).”

In addition to the following information on *envfit* (lines 640-644):

“The envfit function from vegan (78) was used to measure the strength of association between environmental variables and bacterial community data (from the relative abundance table of ASVs). To do so, envfit fitted the environmental vectors (water conductivity, and concentrations of DOC, Al, silicates and chlorophyll a) onto the distance-based CAP ordination.”

Comment: Sometimes bêta-diversity is used instead of beta-diversity

Response: The term was changed to “beta-diversity” at all instances of the revised manuscript.

Reviewer #2 (Comments for the Author):

Comment: In this study by François-Étienne et al, the authors examined the impact of host genotype and environmental factors on the gill microbiome of an Amazonian fish species, *Mesonauta festivus*. The study is really cool and worthwhile, the references cited are appropriate, and the methods used are correct (and thorough!).

Response: Thank you for this comment!

Comment: My main concern is with the writing. The grammar needs to be thoroughly revised throughout the manuscript. In addition, there were some odd word choices that did not really convey what I think the authors were trying for. I gave some examples of recommended changes, especially in the Introduction. I recommend the authors ask a native English speaker to help revise the manuscript.

Response: We thank the reviewer for the recommended changes, which were helpful to revise the language throughout the manuscript. Furthermore, the manuscript was also carefully revised by an additional native English speaker.

Comment: Finally, check the order in which figures appear in the text. The first figure mentioned is Fig. 2.

Response: Corrected.

Comment: Introduction - You mention bacterioplankton only once in the Introduction, despite it being a major component (if not the main component) of your Results and Discussion. I think you need to better introduce and elaborate on the concept. At the very least, you should mention how water type impacts bacterioplankton communities, or discuss what is known about the impact of bacterioplankton on gill microbiomes.

Response: The following paragraph was added to the revised introduction (lines 98-108):

*“Amazonian fish bacterial communities may show a response to the different water types as water physico-chemical parameters are generally known to influence the composition and expression of aquatic microbial communities (23, 24). For instance, a study on the Amazonian tambaqui (*Colossoma macropomum*) detected a high sensitivity of its external bacterial communities to slight variations in environmental physicochemical parameters (25). Furthermore, Amazonian fish bacterial communities could vary according to the differences in the environmental bacterioplanktonic communities, which also differ between water types (26). In other ecosystems, these communities have been known to interfere with host-associated bacterial assemblages via the production of bioactive compounds (27,28), or by modulating the*

assembly process of the eukaryotic host symbiotic community (29-31)."

Comment: Line 28 - "distributed across 12"

Response: Corrected.

Comment: Line 30 - I'm not sure "transcriptionally active" is necessary here. Are there environments on fish where the microbiomes aren't transcriptionally active?

Response: Here, the mention "transcriptionally active" refers to the fact that we used RNA extractions to characterize bacterial communities. Most studies that characterize bacterial communities use DNA extractions, thus detecting both transcriptionally active and dormant/inactive bacteria. A DNA-based approach assumes that all bacteria detected are "active", which is not necessarily the case. The approach used in this study, based on RNA extractions, enabled us to have a more valid representation of the "active" bacteria in the community.

Comment: Line 32 - Replace "in light of" with "using"

Response: Corrected.

Comment: Lines 34-36 - How were they different? Which clusters and treatments were more active?

Response: We agree with the reviewer that it would be interesting to measure the level (i.e. the amount) of transcriptional activity in the different genetic clusters or treatments. However, our study was not designed to do so: Our approach focused on characterizing the relative abundance of the 16S rRNA gene transcripts. It would unfortunately not be possible to speculate on the overall level of transcriptional activity in the different groups based on this data alone. The sentence referred to by the reviewer in this comment was clarified in the revised manuscript to prevent any confusion (lines 34-36):

"Results show that the taxonomic structure of 16S rRNA gene transcripts libraries were significantly different between the four genetic clusters and also between the two water types."

Comment: Line 64 - Replace "diversified" with "diverse"

Response: Corrected.

Comment: Line 65 - Remove "existence of"

Response: Corrected.

Comment: Line 70 - Replace "are characterizing" with "characterize"

Response: Corrected.

Comment: Line 72 - Replace "show a high turbidity" with "are highly turbid"

Response: Corrected.

Comment: Line 77 - What do you mean by "physiologically challenging?"

Response: The following information has been added to the revised manuscript (lines 76-83):

"The relatively high amounts of humic DOC acidify black water environments (pH 3.0-5.0), making them physiologically challenging for local fish (11). For instance, acidic and ion-poor water is known to affect the homeostasis of ionoregulatory processes in different ways (1): Animals living in hypo-osmotic environments generally face the osmotic influx of water and diffusive loss of salts to the external environment (12). In addition, when exposed to acidity (e.g., pH 3.5–5.0) the inhibition of active Na⁺ and Cl⁻ uptake and the elevation of passive ion loss is triggered, resulting in reduced plasma Na⁺ and Cl⁻ levels (13)."

Comment: Line 82 - Replace "evolutive" with "evolutionary"

Response: Corrected.

Comment: Line 83 - Is "global community diversity" the right phrase here? The referenced study focuses on the Amazon.

Response: Corrected to "community diversity" (line 87).

Comment: Lines 90-92 - Some clarification is needed here. Do you mean that the transcriptomes of the microbes are sensitive to variations in environmental parameters, or that the microbial communities are in general sensitive to these variations? If it's the later, it's probably not necessary to specifically compare with the host fish transcriptome.

Response: As suggested by the reviewer, we mean that the "microbial communities are in general sensitive to these variations". To clarify this aspect, the paragraph referred to in this comment has been restructured to the following (lines 85-96):

"Previous studies have examined the effects of these water types on various aspects of Amazonian fish ecology, including phylogeography (14-16), migratory patterns (17), evolutionary strategies (18) and community diversity (19). However, to date only one investigation has studied the effects of black and white waters on native fish bacterial communities (20). Host-associated bacterial communities can often be seen as an extension of the host genome by providing functions critical for the host survival in changing or extreme environments (21). Together with the host (and other non-bacterial microbes), they constitute a holobiont, a term coined by Margulis & Fester (1991) (22), referring to the co-dependence between the host and its microbial symbionts. Measuring to what extent water types affect Amazonian fish bacterial communities is an important first step to understand the physiology of fish-microbe systems in the different Amazonian environments."

Comment: Line 107 - Replace "detritivore" with "detritivorous"

Response: Corrected.

Comment: Line 109 - Replace "They are" with "The species is" or something along those lines. Otherwise, you are alternating between singular and plural.

Response: Corrected to "This species is" (line 123).

Comment: Line 110 - It allows you to test the specific effect of water type, not environmental factors in general.

Response: We agree with the reviewer. This sentence has been corrected to the following (lines 123-125):

"This species is native to both black and white water environments, and thus enables the study of the effect of water type on fish bacterial communities."

Comment: Lines 121-122 - Replace "gill microbiome of wild flag cichlids changes according to the environmental water type and the host genetic cluster" with "gill microbiomes of wild flag cichlids differ among the water types and host genetic clusters"

Response: This sentence has been corrected to the following (lines 136-137):

"Here, our study aimed to characterize how the gill bacterial community of wild flag cichlids differ among the water types and host genetic clusters."

Comment: Line 133 - "specific to the different water types"

Response: Corrected.

Comment: Line 136 - Remove "list of"

Response: Corrected.

Comment: Line 142 - What do you mean by "important"? Do you mean increased?

Response: The word "important" was changed to "increased" in the revised manuscript (line 157).

Comment: Line 143 - Fig. 1 should be listed first. What you refer to as Fig. 1 is in the Methods section which is at the end of the manuscript.

Response: The order of the figures was corrected in the revised manuscript.

Comment: Lines 144-147 - Reword these sentences. Rather say "Firmicutes were significantly enriched in the gill microbiomes of fish collected from black water..." and so on. Also, I would make it more explicit that you are comparing the black and white water microbiomes when you list the statistics, e.g., "12 {plus minus} 5 % in black water compared with 4 {plus minus} 1 % in white water; $p = 0.06$ ".

Response: This section was modified to the following (lines 159-163):

"Firmicutes were significantly enriched in active gill bacterial communities of fish collected from black water (12 ± 5 % in black water compared to 4 ± 1 % in white water, $p = 0.06$). In contrast, Bacteroidetes were significantly enriched in active gill bacterial communities of fish collected from white water (34 ± 4 % in white water compared to 15 ± 2 % in black water, $p = 0.03$)."

Comment: Line 152 - Remove "results from"

Response: Corrected.

Comment: Lines 154-155 - When reporting an F-statistic, you should also report both the numerator and denominator degrees of freedom. One way in which F-statistics are commonly reported is as follows: " $F_{n,97} = 7.49$, $p < 0.001$, $R^2 = 13\%$ " with the degrees of freedom following the F in subscript. See Bowers et al. (2021) for a good example of how to report F-stats.

Bowers, C., Toews, M. D., and Schmidt, J. M. 2021. Winter cover crops shape early-season predator communities and trophic interactions. *Ecosphere* 12(7):e03635

Response: The revised manuscript reported the F-stat as suggested by the reviewer and as in Bowers et al. (2021), see lines 168-171:

"The PERMANOVAs also suggested that the active gill bacterial communities significantly differed according the genetic cluster of the host fish, both in black ($F_{3,97} = 7.49$, $R^2 = 13\%$, $p < 0.001$) and in white water ($F_{3,113} = 10.76$, $R^2 = 14\%$, $p < 0.001$) environments."

And lines 176-179:

"PERMANOVAs confirmed that the active gill bacterial communities significantly differed according the water type of the sampling environment, both in genetic clusters GC2 ($F_{2,57} = 14.30$, $R^2 = 20\%$, $p < 0.001$) and GC4 ($F_{2,37} = 9.33$, $R^2 = 20\%$, $p < 0.001$)."

And at other instances (lines 225, 229, 255, 271, 273).

Comment: Line 166 (and throughout) - I would refer to these as taxa rather than biomarkers.

Response: Although using the word "taxa" would also be appropriate when referring to these bacteria, we believe that the word "biomarkers" helps to differentiate these bacteria from all the other non-biomarker taxa. Ultimately, using this word helps convey our results about genetic cluster- and water type-specific taxa, and it benefits their inclusion in the Discussion (lines 310-326).

Comment: Line 208 - What were the five environmental variables?

Response: This sentence was modified to the following (lines 221-224):

"Permutation tests for CAP under the reduced model showed that the samples significantly clustered according to the five environmental variables provided for the constrained ordination (water conductivity and concentrations of DOC, aluminum, silicate and chlorophyll a), both for bacterioplankton [...]"

Comment: Line 285 - Remove "one of the"

Response: Corrected.

Comment: Lines 317 - Replace "decomposing" with "breaking down", or another synonym.

Response: The word “breaking down” was used as suggested (line 328).

Comment: Fig. 3 - The figure caption needs more details. What do the ovals indicate? Also, the figure text is hard to read and needs to be larger. The statistics shown in the boxes are also listed in the main text, so may not need to be reported in both locations.

Response: The following clarification was added to the figure caption (lines 968-969):

“Ellipses represent groups with default confidence intervals of 0.05.”

Furthermore, the figure text was enlarged and the statistics in the bottom-right corner of each plot were removed as suggested.

Comment: Fig. 4 - This figure is a bit confusing. Fig. 4a and c need a legend. It might be interesting to show how the "biomarkers" in 'a' and 'c' match up with the bar plots in 'b' and 'd' if possible.

Response: A legend was added to the heatmaps in Fig. 3a and c (Fig. 3 was previously Fig. 4 in the original manuscript). In addition, to show how the biomarkers in Fig. 3a and c match with the barplots in Fig. 3b and d, we added a column next to each heatmap (Fig. 3a and c) which shows the taxonomy of the biomarkers represented in the rows of the heatmap.

November 11, 2022

Mx. Francois-Etienne Sylvain
Laval University
Biology
1030 avenue de la Médecine
Quebec, QC
Canada

Re: Spectrum02064-22R1 (Genomic and environmental factors shape the active gill bacterial community of an Amazonian teleost holobiont)

Dear Mx. Francois-Etienne Sylvain:

The response to the the reviewers' concerns were thorough and addressed all of their concerns, therefore I am happy to recommend publishing the article.

Your manuscript has been accepted, and I am forwarding it to the ASM Journals Department for publication. You will be notified when your proofs are ready to be viewed.

Sincerely,

Blaire Steven
Editor, Microbiology Spectrum

Journals Department
Supplemental file 1: Accept